# The Emerging Role of Non-Coding RNAs (ncRNAs) in Plant Growth, Development, and Stress Response Signaling

**DOI:** 10.3390/ncrna10010013

**Published:** 2024-02-07

**Authors:** Amit Yadav, Jyotirmaya Mathan, Arvind Kumar Dubey, Anuradha Singh

**Affiliations:** 1Department of Microbiology & Molecular Genetics, Michigan State University, East Lansing, MI 48824, USA; amitbiot100@gmail.com; 2Sashi Bhusan Rath Government Autonomous Women’s College, Brahmapur 760001, India; jyotirmayabot99@gmail.com; 3Center for Plant Science Innovation, University of Nebraska-Lincoln, Lincoln, NE 68588, USA; arvindbiotech28@gmail.com; 4Department of Plant, Soil and Microbial Science, Michigan State University, East Lansing, MI 48824, USA

**Keywords:** non-coding RNA, root growth, leaf development, seed, endosperm, seed nutrient development, drought, salt, flood, heat, cold stress

## Abstract

Plant species utilize a variety of regulatory mechanisms to ensure sustainable productivity. Within this intricate framework, numerous non-coding RNAs (ncRNAs) play a crucial regulatory role in plant biology, surpassing the essential functions of RNA molecules as messengers, ribosomal, and transfer RNAs. ncRNAs represent an emerging class of regulators, operating directly in the form of small interfering RNAs (siRNAs), microRNAs (miRNAs), long noncoding RNAs (lncRNAs), and circular RNAs (circRNAs). These ncRNAs exert control at various levels, including transcription, post-transcription, translation, and epigenetic. Furthermore, they interact with each other, contributing to a variety of biological processes and mechanisms associated with stress resilience. This review primarily concentrates on the recent advancements in plant ncRNAs, delineating their functions in growth and development across various organs such as root, leaf, seed/endosperm, and seed nutrient development. Additionally, this review broadens its scope by examining the role of ncRNAs in response to environmental stresses such as drought, salt, flood, heat, and cold in plants. This compilation offers updated information and insights to guide the characterization of the potential functions of ncRNAs in plant growth, development, and stress resilience in future research.

## 1. Introduction

Plants exhibit a highly intricate mechanism to synchronize various biological processes and adapt to a spectrum of stress conditions. The gene regulatory mechanism operates at multiple levels, including transcriptional, post-transcriptional, translational, and epigenetic, playing a pivotal role in growth, development, and stress-responsive signaling. This orchestration significantly contributes to the sustainable productivity of plants [1]. Advances in sequencing technologies, tissue-specific and single-cell RNA-sequencing, mass spectrometry-based proteomics, and bioinformatics tools have greatly shifted the focus from gene coding to non-coding elements, maintaining ‘cellular homeostasis’ amid ever-changing environmental conditions [2,3,4]. 

Conventionally, genomic DNA serves as the template for RNA transcription; however, this process does not necessarily result in protein production. The portion of the genome that does not contribute to a gene and hence does not translate into a protein is commonly referred to as “non-coding DNA,” also known as “experimental artifacts” or “junk DNA” [5]. Almost 90% of the genome acts as an RNA template, with only a minor portion coding for messenger RNA (mRNA). Many RNAs are transcribed from the DNA by RNA polymerases and remain functional even if they do not code. These are referred to as non-protein-coding RNAs and are involved in diverse levels of gene regulation, such as epigenetic, transcriptional, and post-transcriptional [6].

Non-protein-coding RNAs, or non-coding RNAs (ncRNAs), constitute a diverse family of RNA molecules transcribed by RNA polymerase but lack codons for protein synthesis [7]. They differ from regular RNAs in their origin and ways of action on target genes and are classified into two categories: regular/housekeeping/constitutive ncRNAs and regulatory ncRNAs (Figure 1). Constitutive ncRNAs, such as ribosomal RNAs (rRNAs), transfer RNAs (tRNAs), small nuclear RNAs (snRNAs), small nucleolar RNAs (snoRNAs), and small Cajal body-specific RNAs (scaRNAs), are constantly expressed in the cells [8]. These ncRNAs typically range from 50 to 200 nucleotides long, facilitating various cellular and ribosomal functions. Furthermore, recent investigations have unveiled complex populations of tRNA-derived small RNA fragments (tRFs) formed by cleaving tRNA at various positions and typically ranging from 17 to 26 nucleotides [9].

Regulatory ncRNAs, on the other hand, are divided into two classes based on their length: long-non-coding RNAs (lncRNAs) exceeding 200 nucleotides and small non-coding RNAs (sRNAs), including microRNAs (miRNAs) of 19–25 nucleotides, small interfering RNAs (siRNAs) of 20–24 nucleotides, and piwi-interacting RNAs (piRNAs) with a length of 24–32 nucleotides [10,11]. The ncRNAs are further categorized based on chromosomal location, sequence/structure, ways of splicing, and biological functions [12,13]. Recently, a unique group of endogenous ncRNAs called “circular” RNAs (circRNAs) with a closed-loop structure has been identified [14,15]. circRNAs are ubiquitous and abundant in all eukaryotes and even in prokaryotic archaea [15]. 

ncRNAs play a wide range of crucial roles in many cellular and biological processes, such as plant growth, development, senescence, and responses to biotic and abiotic stressors, despite their inability to undergo protein translation [10]. These roles are often associated with processes such as RNA splicing, chromatin remodeling, epigenetic memory, turnover, editing, transcription, and post-transcriptional regulation of gene expression [16,17]. The regulatory pathways facilitated by ncRNAs are interconnected, controlling the expression of target genes either synergistically or antagonistically. This intricate regulation protects the genome from both external and internal factors [18]. For example, miRNAs act as negative regulators of target gene expression at the post-transcriptional level, while lncRNAs may compete with miRNAs to build a regulatory network that coordinates the expression of target genes [19,20].

Approximately 35–40 years ago, a single ncRNA was typically identified using a chemical or enzymatic sequencing-based approach. This process involved the size-separation of total RNA on a denaturing gel, followed by visualization and excision of a specific band. In subsequent years, complementary DNA (cDNA) libraries, tilling arrays, and microarrays gained popularity for identifying ncRNAs, each with its own set of limitations and concerns [21]. Recent improvements in computational methods and next-generation sequencing technologies, notably in the realms of transcriptome profiling and single-cell RNA sequencing, have facilitated the identification of thousands of new cell-type-specific ncRNAs. Furthermore, deep learning models are now being developed to predict ncRNA identification and classification [22]. The expression of ncRNAs in the lab can typically be validated using reverse transcription polymerase chain reaction (RT-PCR), and their functionality can be assessed using high-precision single-base CRISPR/Cas9 technology. This allows the creation of ncRNA-related mutants to effectively explore the relationship between ncRNAs and their target genes. 

The first ncRNA discovered in baker’s yeast was alanine-tRNA, which played a pivotal role in understanding genetic regulation. Subsequently, the regulatory role of ncRNAs emerged as a significant revolution in other kingdoms (animals, fungi) as compared to plants [23]. In this review, we emphasized an overview of ncRNAs over the past 20 years (2003–2023) that regulate the target expression of genes and their involvement in growth, development, and stress resilience, contributing to improved agronomic traits for sustainable yield production in many model plant species in the face of future climate change.

## 2. Biogenesis of Housekeeping and Regulatory ncRNAs

### 2.1. Transfer RNA Derived Fragments (tRFs)

tRNAs represent the second most abundant RNA type after rRNAs. They are typically transcribed from precursor tRNA in the nucleus by RNA polymerase III, resulting in the generation of mature tRNA after the removal of 5′ leader and 3′ trailer sequences, as well as the addition of a CCA tail at the 3′ end. The cleavage of 5′ and 3′ ends usually results in transfer RNA-derived fragments (tRF)-5s and tRF-3s, respectively, which range in length from 13 to 20 nucleotides [9]. Mature tRNA with an anticodon loop can be cleaved to produce 5′ or 3′ tRNA halves, spanning in length from 31 to 40 nucleotides. tRFs regulate gene expression through transcription inhibition, RNA degradation, and translation regulation. This regulatory function was evident in processes such as the symbiotic relationship between microbes and soybean roots [24] and the development of pollen grains in Arabidopsis [25].

### 2.2. Small Interfering RNAs (siRNAs)

Small interfering RNAs (siRNAs), typically 21–24 nucleotides long, are ubiquitous in both animal and plant kingdoms. The generation of siRNA is primarily dependent on one of six RNA-dependent RNA polymerases (RDR1–6) that copy single-stranded RNA (ssRNA) to generate double-stranded RNA (dsRNA) [26]. This dsRNA is then precisely processed into small RNA (sRNA) duplexes by RNase III Dicer-like (DCL1-4) proteins. DCL1 predominantly produces 18–21 nucleotide-long sRNA, whereas DCL2, DCL3, and DCL4 generate 22-, 24-, and 21-nucleotide-long sRNA, respectively. After dicing, sRNA duplexes either remain in the nucleus for chromatin-level activities or are exported to the cytoplasm, where they are then assembled into ARGONAUTE (AGO) protein by the RNA-induced Gene Silencing Complex (RISC) for post-transcriptional gene silencing (PTGS) [27]. 

siRNA can originate both exogenously (from viral RNA and transgenes) and endogenously from repeat-rich genomic regions, transposables, and retro-elements [28]. Depending on their origin and processing enzyme involved, endogenous siRNAs are categorized into various types, including cis-acting siRNAs (casiRNAs) or heterochromatic siRNAs (hcsiRNAs; 24 nucleotides), cis-natural antisense-transcript siRNAs (cis-NATs; 24 nucleotides), phased siRNAs (phasiRNAs; 21 or 24 nucleotides), repeat-associated siRNAs (rasiRNAs; 24 nucleotides), and long siRNAs (lsiRNAs; 30–40 nucleotides) [29,30]. 

hcsiRNAs are the most abundant, contributing nearly 70%, and their biogenesis depends on RDR2 and DCL3. They serve an essential function in plant reproduction as well as in maintaining genome integrity and epigenetic modification [31]. cis-NATs, typically present in 6–16%, are usually amplified and processed by RDR6 and DCL1. They also play an important role in plant development and responses to environmental cues [32]. phasiRNAs, a type of secondary siRNA, are typically processed by miRNA-mediated cleavage of RNA transcripts and generated from PHAS loci. These loci, located within the protein-coding region, include many transcription factors (MYBs, NAC), hormone-responsive genes (AUXIN RESPONSE FACTORs; ARFs), and nucleotide-binding leucine-rich repeat (NLR) genes. These ncRNAs function as negative regulators in many biological processes [33]. In addition, PHAS loci are found within noncoding regions, producing lncRNAs, which then generate trans-acting siRNAs (tasiRNAs) through TAS loci [34]. tasiRNAs function in a *trans* manner and are involved in post-transcriptional regulation, particularly in anther development [35]. rasiRNAs, derived from transposons and repetitive genomic regions, mediate transcriptional gene silencing to maintain genome stability. Distinct rasiRNAs were discovered between two maize-inbred lines (B73 and Mo17) as well as their hybrid, contributing to a high degree of heterosis [36]. lsiRNAs, usually processed by DCL1, are induced either in response to pathogen infection or under specific growth conditions. For instance, in Arabidopsis, AtlsiRNA-1 to AtlsiRNA-5 was induced by *Pseudomonas syringae* carrying effector *avrRpt2,* which silences RAP-domain proteins involved in disease resistance [37]. 

### 2.3. MicroRNAs (miRNAs)

Plant microRNAs (miRNAs) are endogenous ncRNAs present in a diverse range of organisms, including both animal and plant kingdoms. They typically consist of approximately 20–24 nucleotides. The first miRNA was discovered in *Caenorhabditis elegans*, where it orchestrates developmental timing by binding to specific target mRNAs [38]. 

miRNAs typically originate from a stem-loop structure and can be classified as either ‘intergenic’ or ‘intronic’. They are transcribed by miRNA (MIR) genes through RNA polymerase II (RNAPII), resulting in single-stranded primary miRNA transcripts (pri-miRNAs). Subsequently, aided by HYPONASTIC LEAVES 1 and SERRATE, pri-miRNA undergoes two cleavage events and is processed into a double-stranded miRNA through DICER-LIKE (DCL1). The processed miRNA is then incorporated into the AGO1 protein, exported to the cytoplasm, and binds to target mRNAs. This initiates PTGS by assembling the miRNA-induced silencing complex (miRISC) [39]. The AGO4-mediated RISC usually generates a 24-nucleotide miRNA and is involved in epigenetic modification by targeting DNA methylation at the transcriptional level [34]. 

miRNAs typically inhibit the expression of target transcripts via two main mechanisms: transcript cleavage and translational inhibition. Specifically, transcript cleavage is facilitated by exoribonuclease activity, while translation inhibition happens through targeting the 5′ untranslated region to prevent ribosome recruitment [40]. Notably, a type of peptide known as miRNA-encoded peptides (miPEPs) is translated from short open reading frames (sORFs) in the 5′ leader sequence of pri-miRNAs [41,42]. These peptides promote the transcription of their corresponding pri-miRNAs, thus regulating many biological processes [43]. For instance, miPEP165a and miPEP171b suppress the target genes associated with lateral root development and stimulate main root growth in *Arabidopsis* and *Medicago*, respectively [43]. Furthermore, vvi-miPEP171d1 has been shown to regulate adventitious root formation in grapevine [44], while miPEP858a altered plant development and induced secondary metabolite accumulation by influencing the expression of genes involved in phenylpropanoid pathways and auxin signaling in Arabidopsis [45].

MIR genes are intergenic and rarely arranged in tandem, although clustering does not seem uncommon in some plants, such as soybeans [46]. As of 5 February 2024, nearly 326 MIR gene families have been deposited in the miRbase registry for *Arabidopsis thaliana*, 174 for *Zea mays*, 122 for *Triticum aestivum*, 604 for *Oryza sativa*, 247 for *Physcomitrella patens*, and 58 for *Selaginella moellendorffii* (http://microrna.sanger.ac.uk/sequences, accessed on 5 February 2024) [26]. Interestingly, in Arabidopsis, nearly 100 MIR gene families represent an evolutionary fluid set of molecules that arose after the split between land plants and mosses but before the monocot/dicot divergence. Some miRNA families are conserved in mosses, indicating a very ancient origin. These include miR156, miR160, miR319, and miR390, all of which regulate ancestral transcription factors responsible for specifying basic meristem functions, organ polarity and separation, cell division, and hormonal control [47,48]. 

### 2.4. Long Non-Coding RNAs (lncRNAs)

Long non-coding RNAs (lncRNAs) are another type of ncRNA that exceed 200 nucleotides in length and lack discernible coding potential. They are found ubiquitously in plants, animals, fungi, and prokaryotes. Although most lncRNAs are primarily located in the nucleus and associate with chromatin, they can function in both nuclear and cytoplasmic compartments [49]. lncRNA can be broadly classified into three categories: long intergenic RNAs (lincRNAs), intronic ncRNAs (incRNAs), and natural antisense transcripts (NATs) [11]. incRNAs are transcribed in any orientation relative to coding genes, while NATs are transcribed from the antisense strand of genes. 

Eukaryotes deploy three conserved multi-subunit RNA polymerases (Pol I, II, and III) to transcribe their nuclear genomes into various coding and noncoding transcripts [11]. All three of these RNA polymerases contribute to lncRNA production. Pol I and Pol III are specifically dedicated to lncRNA production, while Pol II, despite its primary function in mRNA biogenesis, also produces a range of lncRNAs. Notably, in plants, two additional enzymes, Pol IV and Pol V, are utilized to produce lncRNAs, which play a crucial role in recognizing and silencing transposable elements [50].

In terms of gene regulation, lncRNAs exhibit diverse mechanisms, operating at multiple levels through both simple and complex mechanisms. They can function in *cis* or *trans*, operate through sequence complementarity to RNA or DNA, and are recognized via specific sequence motifs or secondary/tertiary structures [51]. Furthermore, lncRNAs may compete with mRNAs for miRNA molecules, regulating miRNA-mediated target repression through competing endogenous RNA (ceRNA) [52]. Overall, lncRNAs can serve as precursors of miRNAs and siRNAs and act as endogenous target mimics (eTM), competing for various miRNAs [53]. 

In recent years, the rapid development of various technologies has led to the discovery of different types of lncRNAs. The first identified plant lncRNA, *ENOD40,* was isolated from alfalfa and demonstrated to modify the subcellular localization of the nuclear RNA-binding protein MtRBP1 (RNA-binding protein 1) [54,55]. Subsequently, several lncRNAs involved in many developmental and stress resilience processes have been identified. For instance, lncRNA SVALKA is associated with freezing response [56], seed dormancy through the Delay of Germination 1 gene (*DOG1 antisense*) [57], and *FLOWERING LOCUS C* (FLC) gene silencing during vernalization through *COLD ASSISTED INTRONIC NONCODING RNA* (*COLDAIR*) [58], *COLDWRAP* [59], COLD INDUCED LONG ANTISENSE INTRAGENIC RNAs (*COOLAIR*) [60], and *COLD INDUCED lncRNA 1* (*CIL1)* [61].

Many sORFs, like miPEPs, are embedded in lncRNAs, encoding functional micropeptides. These can be predicted using several bioinformatics tools, such as FuncPEP, SmProt, PsORF, and many more [62], and their existence can be confirmed through ribosome profiling and mass spectrometry methods [2]. Notably, micropeptides such as DEVIL1 [63], ROTUNDIFOLIA [64], and POLARIS [65] have been identified in Arabidopsis, which play an important role in organ development, leaf morphogenesis, and root growth, respectively. Additionally, Lin et al. [66] identified a subset of micropeptides involved in metabolite, energy, and defense-related processes in soybean. 

Presently, single-cell RNA sequencing has been extensively used to identify cell-type-specific lncRNAs and lncRNA-associated gene regulatory networks [67]. Recently, He et al. [68] identified cell-type-specific lncRNA signatures from Arabidopsis seedlings and deposited all the data at the single-cell-based plant lncRNA atlas database (scPLAD), which can be used for functional dissection of lncRNAs in plants. 

### 2.5. Circular RNAs (circRNAs)

Over the past decade, a multitude of circular RNAs (circRNAs) have been identified, demonstrating their ubiquity and abundance across all eukaryotes [14,15,69,70]. circRNAs are single-stranded, endogenous RNAs characterized by a distinct closed-loop structure. They are produced using a non-canonical process known as “back-splicing,” in which the 5′ and 3′ ends are linked by a covalent bond. Due to their closed-loop structure, circRNAs exhibit greater stability compared to linear RNAs and are less susceptible to degradation by Ribonuclease R (RNase R). Although back-splicing is generally less efficient than linear splicing and their expression levels are relatively low, circRNAs can accumulate in specific cell types or tissues in a temporally regulated manner owing to their high stability [71]. Usually, rRNA-depleted total RNA-seq is widely used in early genome-wide profiling studies of circRNAs. However, due to low expression levels of circRNAs, an additional protocol that uses linear RNase R for library preparation has become the preferred method for greatly enriching circRNAs [72,73]. 

The precise biogenesis of circRNAs remains unclear, but two conserved models have been proposed to explain their formation. In the first model, circRNAs typically originate from canonical splicing sites, facilitated by base pairing between inverted repeat elements located in upstream and downstream introns or through the dimerization of RNA-binding proteins (RBPs) binding to specific motifs in the flanking introns [15]. In the second model, circRNAs are produced from lariat precursors during exon-skipping or from intron lariats that escape debranching [74].

circRNAs can arise from a wide range of genomic positions and combinations, including exons (exonic circRNAs), introns (intronic circRNAs), intergenic regions (intergenic circRNAs), or UTR circRNAs [75]. circRNAs are involved in the regulation of gene expression at transcriptional and post-transcriptional levels. They are reported to function as miRNA sponges [76] or RBP sponges [77] and are considered ideal biomarkers [78]. 

### 2.6. piwi-Interacting RNAs (piRNAs)

piRNA, or piwi-interacting RNA, was first discovered in *Drosophila melanogaster*, derived from single-stranded RNA and exclusively present in animals. Mature piRNAs are around 21–34 nucleotides in length, with a preference for uridine at the 5′ end and 2′ *O*-methylation at the 3′ end [79]. piRNAs bind and guide PIWI proteins to specific mRNA targets through sequence complementarity between the target RNA and piRNAs. They play important roles in gene regulation, transposon element repression, and antiviral defense [80]. Since piRNAs are not found in plants, they will not be the focus of this review article.

## 3. Role of ncRNAs in Plant Growth and Development

Plant ncRNAs have emerged as important regulators of chromatin dynamics, exerting influence over transcriptional programs and contributing to various developmental processes such as root, leaf, flower, endosperm, and seed development. These ncRNAs are pivotal in shaping the epigenetic landscape and coordinating gene expression to achieve specific developmental outcomes in plants (Table 1).

### 3.1. Root Growth and Development

Roots are indispensable plant organs lying beneath the soil surface that not only provide essential structural support to the aerial portion of the plant but also play a vital role in acquiring nutrients and water for growth. In Arabidopsis, the root system architecture comprises primary roots (PRs) and individual lateral roots (LRs) [81]. PRs develop from stem cells surrounding the quiescent center through several rounds of asymmetrical cell divisions and expansions, eventually forming the root meristem [82]. Conversely, LRs originate from the initial specification of pericycle cells, a heterogeneous tissue composed of quiescent cells at the phloem poles and specified cells designated to form lateral root primordia opposite each xylem pole [83]. 

Several classes of ncRNAs, including siRNAs [84,85], miRNAs [86,87,88], lncRNAs [89,90,91,92,93,94], and circRNAs [95], have been identified as important regulators of root development in many plant species (Table 1). Among those studies, nuclear speckle RNA-binding proteins have been implicated in auxin-regulated LR formation and function as regulators of alternative splicing (AS) [93]. These proteins interact not only with some of their alternatively spliced mRNA targets but also engage with Alternative Splicing Competitor RNA (ASCO), a structured lncRNA comprising *ENOD40* and lnc351 [90]. This interaction allows them to influence gene expression by hijacking AS regulators [92].

Root hairs (RHs) are specialized single-cell projections that originate from epidermal trichoblast cells [96]. They play essential roles in absorbing water-soluble nutrients, interacting with soil microorganisms, and anchoring the plant. Recently, the AUXIN-REGULATED PROMOTER LOOP (APOLO) lncRNA has been shown to recognize multiple spatially independent genes by sequence complementarity and DNA–RNA duplex formation, known as R-loops [89]. APOLO integrates both endogenous (PINOID) and exogenous signals, working in conjunction with a complex network of epigenetic factors such as the polycomb repressive complex 1 (PRC1) component LIKE HETEROCHROMATIN PROTEIN 1 (LHP1), which maintains a repressive chromatin status [89]. APOLO lncRNA controls the master regulator of RHs, ROOT HAIR DEFECTIVE 6 (RHD6) at low temperature (10 °C) by decoying LHP1 away from chromatin through interaction with WRKY42 [94]. This modulates WRKY42 binding to the RHD6 promoter, leading to the activation of downstream target genes ROOT HAIR DEFECTIVE SIX-LIKE 2 (RSL2) and ROOT HAIR DEFECTIVE SIX-LIKE 4 (RSL4), which promote RH growth and development [91]. 

The formation and development of adventitious roots (ARs) are crucial for the vegetative propagation of woody species, particularly in poplar [97]. ARs originated from a cell layer reminiscent of the root pericycle, sharing histological and developmental characteristics with LRs [98]. Plant-specific WUSCHEL-related homeobox (WOX) transcription factors (TFs), especially WOX11, LATERAL ORGAN BOUNDARIES DOMAIN 16, and SMALL AUXIN-UP RNA 36 (SAUR36), are essential regulators of AR formation via the auxin pathway [99]. Numerous lncRNAs that regulate the expression of WOX11 have been identified [97,100]. Among them, lncWOX11a is located 54,293 bp upstream of *PeWOX11a* on chromosome 13 and is mainly expressed in 1-week-old roots. Overexpression of lncWOX11a inhibits AR development, while CRISPR/Cas9 knockout demonstrated that lncWOX11a acts as a negative regulator of adventitious rooting by downregulating PeWOX11a [97]. 

### 3.2. Leaf Development

Leaf development is an intricate and sequential process that plays a critical role in plant growth. Leaves, as lateral organs, go through various key developmental stages, including leaf primordium initiation (arising from the shoot apical meristem), leaf polarity establishment, development phase transition, leaf morphology modulation, and eventually leaf senescence [101,102]. 

In the realm of leaf development, multiple siRNAs [103,104,105], miRNAs [47,106,107,108,109], lncRNAs [110,111], circRNAs [112], and miRNA-transcription factor regulatory modules [113,114,115,116] have also been identified, each playing pivotal roles (Table 1). For example, miR164 negatively regulates the CUP-SHAPED COTYLEDON (CUC) genes, CUC1 and CUC2, which encode a paralogous pair of NAC transcription factor and are implicated in leaf serration [115]. Similarly, miR165/166 targets the class III homeodomain leucine zipper transcription factor (HD-ZIPIII), crucial for regulating processes such as meristem initiation, lateral organ polarity, and vascular development [117,118,119]. In Arabidopsis, the miR165/166 gene family comprises nine members, including two miR165 (miR165a and miR165b) and seven miR166 (miR166a to miR166g) [120]. Despite a single nucleotide difference between the mature microRNAs of these two families, their target sites within the HD-ZIPIII are highly conserved across all land plants. Notably, the expression patterns are complementary, with HD-ZIPIII transcription factor expressed in the meristem and on the adaxial side of developing leaves, while miR166 accumulates on the abaxial surface of young leaf primordia in Arabidopsis [118].

The evolutionarily conserved miR319 regulates TCP, a plant-specific transcription factor named after TEOSINTE BRANCHED, CYCLOIDEA, and PCF1/PCF2 [109]. miR319 affects TCP2, TCP3, TCP4, TCP10, and TCP24, as well as their homologs in different plant species [106]. Elevated levels of miR319 lead to the downregulation of these TCPs, resulting in dramatic alteration in Arabidopsis leaf morphogenesis and the development of crinkled leaves [121,122,123]. miR396 targets seven genes (GRF1-4 and GRF7-9) from the growth-regulating factor (GRF) [124,125]. Mutations in various GRFs or overexpression of miR396 result in reduced leaf size, indicating partial functional redundancy [114]. Conversely, overexpressing GRFs or mutating the miR396-binding site of GRF2, GRF7, or GRF9 results in slightly larger leaves [125,126].

As plants reach the maturation stage, leaves undergo organ-level senescence, which is a critical process for optimal offspring production and plant survival. A recent study by Kim et al. [111] conducted comprehensive RNA-seq analyses on Arabidopsis leaves, representing all developmental stages. This led to the identification of approximately 770 age-related (AR) lnRNAs. Among them, AT1G33415 and AT2G14878 were shown to be coexpressed with Autophagy 9 and ADENINE PHOSPHORIBOSYL TRANSFERASE 1, regulating leaf senescence through decreased cytokinin metabolism [127]. Furthermore, three competitive endogenous RNAs (ceRNAs) were identified, which involved lncRNA-miRNA-mRNA interactions. The AR-lncRNAs AT4G36648, AT5G23410, and AT1G26208 interact with miR164, miR169, and miR156 to target ORESARA1, NUCLEAR FACTOR Y (NF-Y), and SQUAMOSA PROMOTER BINDING-LIKE (SPL), respectively [111]. This regulation is crucial for age-dependent developmental transitions, specifically in leaf senescence, as well as responses to abiotic and biotic stresses [128]. Overall, these studies shed light on the intricate miRNA-TF regulatory network governing leaf development and offer potential avenues for manipulating leaf-related traits in plant species.

### 3.3. Seed, Endosperm, and Seed Nutrient Development

Grain development constitutes a pivotal biological process intricately connected to both grain yield and quality in plants. The complexity of grain yield as a trait is determined by several factors, including spike/panicle number per plant, grain number per spike/panicle, and one thousand kernel weights. Notably, these factors are predominantly influenced by the architecture of the spike/panicle [129]. 

Several ncRNAs [130,131,132,133,134,135,136,137] have been identified that regulate seed-related features, as summarized in Table 1. Notably, a comprehensive study by Zhang et al. [136] discovered nearly 65 lncRNAs associated with rice reproduction using the strand-specific RNA sequencing method. Among them, three lncRNAs (XLOC_0063639, XLOC_007072, and XLOC_057324) were highly expressed during the reproductive stage. Another study by He et al. [138] identified a quantitative trait locus linked to improved rice yield, comprising several tandemly arranged intronless leucine-rich repeat receptor kinase (LRK) genes. Subsequently, Wang et al. [134] identified lncRNA transcribed from the antisense strand of the neighboring gene LRK cluster, named LRK Antisense Intergenic RNA (LAIR). Silencing LAIR in transgenic lines resulted in inhibited growth and reduced expression levels for all LRK genes. Conversely, lines overexpressing LAIR exhibited significant increases in total grain yield, coupled with enhanced expression of some members of the LRK gene cluster. 

In wheat, Cao et al. [137] also identified 2753 differentially expressed lncRNAs, highlighting their significant roles in spike development. However, this finding was limited to a single wheat genotype, leaving the genetic variations and evolutionary characteristics of wheat lncRNAs largely unexplored. Yang et al. [139] addressed this gap by conducting a comprehensive study involving 93 wheat genotypes to unveil the lncRNA landscape associated with wheat spikes. The analysis yielded a total of 35,913 lncRNAs, along with 1619 lncRNA-mRNA pairs. Among these pairs, lncRNA.127690.1-TraesCS2A02G518500.1 and lncRNA.104854.1-TraesCS6A02G050300 were identified as regulators of heading date and spike length, acting through pectin methylesterase inhibitor protein and autophagy protein (AtATG5), respectively.

Oil is a valuable component in seeds that plays an important role in human health [140]. Most oilseed crops produce and accumulate oil during seed development and maturation, involving morphogenesis, cell enlargement/reserve accumulation, and desiccation or developmental arrest [141]. Oil biosynthesis in seeds occurs in two subcellular compartments and involves four key steps: de novo fatty acid synthesis, acyl editing, triacylglycerol assembly, and oil drop formation [141]. Recently, lncRNAs have been implicated in regulating seed oil biosynthesis [142,143,144,145]. For instance, in the developing seeds of *Brassica napus*, the expression patterns of 13 lncRNAs were significantly correlated with lipid-related genes, suggesting potential roles for these lncRNAs in oil synthesis [142]. Subsequently, Li et al. [145] identified 8094 expressed lncRNAs across three developing stages in two accessions of *Brassica napus*, ZS11 (high oil content) and WH5557 (low oil content). Notably, lncRNAs MSTRG.22563 and MSTRG.86004 were predicted to be associated with seed oil accumulation. Overexpression of MSTRG.22563 reduced seed oil content by 3.1–3.9% and altered the fatty acid composition of seeds. In contrast, the lncRNA MSTRG.86004 boosted oil content by 2.0% while delaying seed development. 

Endosperm development is a crucial process in cereal grains that directly impacts human nutrition. It is an important tissue for studying the molecular mechanisms that drive its development as well as metabolic processes contributing to the formation of its storage constituents. The development sequence initiates with an initial syncytial phase, followed by cellularization, in which the endosperm nuclei undergo multiple rounds of mitosis without cytokinesis, resulting in a multinucleate cell [146]. The transition from syncytium to cellularization is pivotal for determining final seed size, making it a useful model for studying the cell cycle and cytokinesis. The helicase family protein (HeFP) plays a vital role in endosperm cellularization [147]. Hara et al. [148] identified the rice SNF2 family helicase ENDOSPERMLESS 1, which regulates syncytial endosperm development. In a subsequent study, Zhou et al. [149] uncovered a mutant with a T-DNA insertion in a lncRNA, *MIS-SHAPEN ENDOSPERM* (*MISSEN*; XLOC_057324), maternally expressed and repressed by histone H3 lysine 27 trimethylation (H3K27me3) post-pollination. *MISSEN* suppressed endosperm development by negatively regulating HeFP, leading to a prominent dent and bulge in the seed. Both studies revealed the potential applications of the HeFP pathway in breeding for improved grain yield in rice.

The endosperm is primarily made up of starch, which forms a complex glucose polymer with a linear chain of amylose and highly branched amylopectin [150]. The dietary consumption of cereals, particularly those with high glycemic content, is often associated with diet-related diseases such as obesity, type-2 diabetes, and various cardiovascular diseases. Elevating amylose content contributes to type 2 resistant starch (RS), as it undergoes slow digestion in the intestine, resulting in a low glycemic index [151]. In a recent study, Madhawan et al. [152] uncovered a lncRNA–miRNA–mRNA regulatory network where TCONS_00130663 interacts with miR1128 to regulate lysophospholipase gene expression. Concurrently, this lncRNA, which has a negative correlation with starch branching enzyme IIb (SBEIIb) in a high RS mutant line, may have a role in regulating the RS via an unknown mechanism. In another study, Zhang et al. [153] also identified several lncRNAs, which coexpressed with genes involved in starch metabolism such as *Brittle 1* (TraesLNC6D200.1; *TaBT1-6D*), cell wall invertase genes (*TaCwi-5D;* TraesLNC5D5400.1 and *TaCwi-4A;* TraesLNC4A35700.1), sucrose transporter gene (*TaSUT-1A;* TraesLNC1A1200.1), sucrose non-fermenting 1-related protein kinase 2 (*TaSnRK2.10-4A;* TraesLNC4A37800.1), and granule bound starch synthase II (*TaGBSSII;* TraesLNC2B15700.1). These lncRNAs not only regulate starch metabolism but also serve as potential candidates for functional studies on wheat grain size and weight regulation.

Storage protein (SP), which includes glutelin and gliadin, represents another important component in the endosperm that not only determines the baking quality but also contains several important amino acids such as proline, glutamine, asparagine, and arginine [154]. Several lncRNAs have been found to coexpress with SP in wheat [153,155]. For instance, the antisense lncRNA TraesLNC1D26001.1, which is transcribed from the promoter of the high molecular weight glutelin subunit (HMW-GS), coexpressed strongly with *TaGLU-1Dy.* Functional analysis of TraesLNC1D26001.1 revealed no significant changes in protein level, suggesting that lncRNA does not regulate the transcription or translation of its adjacent gene. However, the study identified *ABSCISIC ACID-INSENSITIVE 5* (*AIB5*), a putatively trans-regulated gene involved in seed maturation, germination, and abscisic acid (ABA) response, showing higher expression in lncRNA-overexpressed constructs [155]. The *AIB5* further exhibited reduced germination compared to the wild type, especially with increased ABA construction, indicating that TraesLNC1D26001.1 overexpression enhanced ABA sensitivity during seed germination [155]. The delay of seed germination was also associated with a seed vigor-related lncRNA (SVR; lncRNA MSTRG.182510.6) that interacts with members of the SAUR gene family. CRISPR/Cas9-engineered mutations in SVR delayed rice germination through the auxin-activated signaling pathway [156]. Overall, this research provides valuable insights into the regulatory role of lncRNAs in seed-related biological processes, offering potential applications in crop improvement.

## 4. Role of ncRNAs in Various Abiotic Stress Resilience

Plants, as sessile organisms, are exposed to a variety of abiotic stresses, including heat, drought, salinity, flooding, low temperatures, and others. These stresses can lead to a substantial reduction in plant growth and overall productivity, with documented impacts ranging from 50 to 70% worldwide [157,158,159]. Abiotic stresses, especially those linked to global warming, have a substantial and pervasive deleterious impact on plants on a broad geographic scale. The escalating prevalence of these stresses, exacerbated by factors including climate change, poses significant challenges to global agriculture and plant ecosystems. To effectively handle this scenario, dedicated research, the development of stress-tolerant crop varieties, and the implementation of sustainable agricultural practices are essential. These approaches aim to reduce the negative impacts of abiotic stresses on plant productivity while increasing their resilience in the face of environmental challenges. 

### 4.1. Drought Stress

Drought stress poses a major agricultural risk, prompting urgent global efforts to enhance drought-tolerant crops. Drought stress has a deleterious impact on plants, including osmotic imbalance, membrane system degradation, reduced respiration, and photosynthetic rates. These responses not only impede plant growth and metabolism throughout its life cycle but also diminish crop yield and quality. 

A list of several drought-responsive ncRNAs has been summarized by Gelaw et al. [160], which are listed in Table 1 [161,162,163,164,165,166,167,168,169,170,171,172,173]. Most of them target genes involved in ethylene and ABA synthesis or signaling, calcium signaling, starch and sucrose synthesis, and a variety of other metabolic processes. For instance, the lncRNAs present in rice, NAT Os02g0250700-01 and NAT Os02g0180800-01, target the late embryogenesis abundant protein and cinnamoyl CoA reductase genes, respectively [168]. Numerous lncRNAs in maize were discovered from three distinct tissues (ear, tassel, and leaf) across four developmental stages (V12, V14, V18, and R1) under both well-watered and drought conditions [169,170]. Among the total 3,488 high-confidence identified lncRNAs, MSTRG.2834.1 responded to drought specifically in the ear, with the greatest expression changes at the R1 stage and negligible changes at the V12, V14, and V18 stages upon drought stress. However, drought stress barely influenced the expression of lncRNA MSTRG.43642.1 in the ear at the V12, V14, and V18 stages, but it was significantly downregulated at the R1 stage. They also identified 1177 lncRNA-mRNA interactions, with lncRNA MSTRG.6838.1 targeting vpp4 (GRMZM2G028432), encoding a vacuolar (H+)-pumping ATPase subunit gene. The expressions of vpp4 and MSTRG.6838.1 were significantly correlated in various tissues and development stages, and they were both down-regulated under drought stress, indicating that lncRNA MSTRG.6838.1 and vpp4 may be promising cis-acting pairs that hold potential for co-regulation. 

The discovery of drought-responsive lncRNA (DRIR) demonstrates its crucial role in regulating plant tolerance to drought and salt stresses [171,172]. Zhang et al. [173] recently conducted large-scale profiling and identified 2174 and 1354 high-confidence circRNAs in maize and Arabidopsis, respectively. Specifically, they found two circRNAs associated with the GRMZM2G032852 and GRMZM5G817173 genes, which encode a calcium-dependent protein kinase (CPK) and cytokinin oxidase/dehydrogenase (CKX), respectively. These circRNAs, named circCPK and circCPK, were formed through back-splicing between exons, resulting in the insertion and deletion of alleles in the maize panel. The expression levels of circCPK and the survival rate of maize plants with the insertion alleles were notably higher than those with the deletion alleles under both normal and drought conditions. This observation suggests a potential association between circRNA expression and plant drought tolerance. 

Furthermore, in another study, the researchers investigated the role of circGORK, which is hosted by the GORK (Guard cell outward-rectifying K+-channel) gene [174]. GORK encodes a potassium output channel that is critical for the regulation of ABA signaling and water stress responses [175]. Overexpression of circGORK in transgenic Arabidopsis plants had a negative impact on the ABA response. Specifically, the germination rate of transgenic seeds was considerably lower on ABA plates than that of control seeds, suggesting that circGORK may modulate ABA signaling pathways and influence water stress responses.

### 4.2. Salt Stress

Salt stress poses a substantial barrier to plant growth and development, affecting approximately 33% of arable land. It increases both ionic and intracellular osmotic pressures in plants, resulting in increased toxicity, oxidative damage, and osmotic stress [176]. Rice is vulnerable to salt stress, with most cultivated varieties having an electrical conductivity threshold of 3 dSm^−1^ [177]. 

Limited evidence is available on ncRNAs associated with the regulation of salt stress in plants [177,178,179,180,181,182,183,184,185,186,187,188,189,190], as listed in Table 1. Nearly 75 conserved miRNAs were identified from the control and NaCl-treated salt-tolerant rice varieties [180]. Among them, osa-miR12477, which targets L-ascorbate oxidase, showed an accumulation of oxidative stress in the plant upon salt treatment. Similarly, Jain et al. [191] identified 1500 and 2100 putative lncRNAs from the root and leaf tissue of salt-tolerant (Horkuch) and salt-sensitive (IR-29) rice genotypes, respectively. The study further identified two differentially expressed (DE)- lncRNAs, such as TCONS_00008914 and TCONS_00008749, as potential target mimics of known rice miRNAs. In another study, using another salt-tolerant rice (FL478), Mansuri et al. [177,187] identified DE-lncRNA.2-FL, that showed multiple interaction with osa-miR2926 miRNA. This interaction regulates the expression of various salt-stress-responsive genes, such as chloride channel protein, potassium transporter, and some genes involved in sensing and signaling, such as cysteine-rich receptor-like protein kinase 8 precursors and serine/threonine protein kinase. The lncRNA also identified pentatricopeptide repeat protein as a *cis*-target and multiple *trans*-targets that mediate regulatory role in a salt-tolerant rice variety. 

lncRNA DRIR was identified as a positive regulator of the plant response to salt stress, in addition to its role in drought stress [171]. Under normal conditions, DRIR expression was modest, but it was significantly induced by drought, salt stress, and ABA treatment. A genome-wide transcriptome analysis was conducted to investigate the impact of DRIR on gene expression in response to stress. The results indicated changes in the expression of genes involved in ABA signaling, water transport, and other stress-relief processes in both the *drir* mutant and overexpressing plants. The study provides valuable insights into the regulatory role of DRIR in the complex network of molecular responses that plants undergo to cope with drought and salt stress. 

### 4.3. Flooding/Waterlogging/Hypoxia Stress

Flooding is the temporary submersion of land by water, while waterlogging happens when the soil is either fully saturated or near saturation most of the time. Both processes hinder various physiological processes in plants, such as photosynthesis, disturb cellular energy production, and disrupt ionic balance [192]. These effects worsen when plants encounter lower oxygen content below optimal conditions, a phenomenon known as hypoxia, which frequently happens in the natural environment of plants. The restricted availability of oxygen in the root zone adversely impacts the overall metabolism of the plant, impairing its growth and development [193].

Several ncRNAs have been detected in different plant species exposed to waterlogging or hypoxia conditions, such as Arabidopsis [194], maize [195,196], rice [197], Populus [198], Medicago [199], tomato [200], and cucumber [201,202], and some of them are shown in Table 1. For instance, in rice, nearly 291 and 58 miRNAs responsive to deepwater stress were discovered in the internode and node, respectively [197]. Further, degradome analysis predicted nearly 1586 and 744 targets cleaved by 230 and 53 miRNAs from the internode and nodes, respectively. Notably, miR156 targeted the SQUAMOSA-promoter binding protein (SBP)-box gene (cleaved at 1163 nts), and NF-Y transcription factor (cleaved at 1729 nts) was targeted by miR169. The study further identified miRNA-lncRNA-mRNA regulatory networks, such as nov-miR145-TCONS_00011544 and miR159e-TCONS_00059798, targeting ZFD (Os11g36430) and MYB (Os01g36460) genes, respectively.

Flooding is also associated with reduced energy metabolism, as detected by a sucrose non-fermenting-related protein kinase (SnRK1) [203]. Overexpression of miR156 has previously been shown to improve abiotic stress tolerance in various plants via regulating SQUAMOSA-promoter binding protein-like (SPL) transcription factors [204,205]. Later, Feyissa et al. [199] studied the role and putative mode of action of the miR156/SPL module in alfalfa flooding tolerance. They utilized many genetic constructs, including miR156 overexpression (miR156OE), SPL6RNAi, SPL13_RNAi, wild-type, and empty vector plants, along with flood-tolerant (AAC-Trueman) and flood-sensitive (AC-Caribou) alfalfa cultivars. The study revealed that the miR156/SPL module mediates physiological responses in alfalfa under flooding conditions. Specifically, in miR156OE alfalfa plants, SnRK1 expression increased in an ABA-dependent manner. SnRK1 is a key regulator of energy and stress signals in plants. The findings suggest that the miR156/SPL module contributes to flooding tolerance by regulating physiological processes and SnRK1 expression, highlighting the intricate regulatory mechanisms involved in plant stress responses [199]. 

### 4.4. Heat and Cold Stress

Both high and low temperatures (chilling at 0–15 °C and freezing at <0 °C) impact plant growth, development, and various physiological functions. Numerous temperature-responsive regulatory genes, such as heat shock factors and C-repeat/dehydration-responsive element binding factors (CBF/DREB1s), bind to the promoters of heat stress response genes and cold-regulated genes (COR), thereby promoting plant adaptability to heat and cold stress, respectively [206]. 

An array of heat and cold stress-responsive ncRNAs have been identified, comprising several siRNAs [207,208,209], circRNAs [210,211,212], and miRNA-targets [213,214,215,216,217,218,219,220,221,222,223,224,225,226,227,228,229]. Notable examples of miRNAs and their targets that are linked to either heat and cold stress include but are not limited to: miR156-SPL [213,214,215], miR159-GAMYB [216], miR160-ARF [217], miR164-NAC [218], miR165/166-homeodomain leucine zipper class-III transcription factor mainly PHBULOSA and ATHB-8 [219], miR167-ARF6 [220], miR168-AGO1 [221], miR171-GRAS gene family [222], miR172-APETALA2 [223], miR319-TCP [224,225,226], miR393-auxin receptors with F-box domains; TIR1/AFB2 [227], miR396-GRF [228], miR397-laccase gene; lignin-degrading enzyme [222], miR402-DEMETER-LIKE PROTEIN3; a 5-methylcytosine DNA glycosidase [229]. Genome-wide transcriptomic analyses have further identified conserved miRNA (miR166) and unique miRNAs (miR168, miR444, and miR528) in the maize leaves during the V3 stage under heat stress [230]. In sweet potato seedlings, specific miRNAs such as IbmiR162, IbmiR164, IbmiR171, and IbmiR397 have been implicated in physiological changes, particularly oxidative stress during chilling and heat stress [231].

Several lncRNAs were also identified for both heat and cold stress using chip-based microarray and strand-specific sequencing approaches, with high specificity in tissue types and developmental stages [232,233,234,235,236,237]. For instance, lnc-173 targets the SUCROSE SYNTHASE4 gene, and under heat stress, only the target gene was expressed, not the lncRNA itself [237]. Heat shock transcription factors (HSFs) play a vital role in the response and acclimation of eukaryotes under heat stress. In Arabidopsis, asHSFB2a (natural antisense transcript of HSFB2a) was found to be induced by heat stress and negatively regulate HSFB2a expression [238]. Recently, a *Heat-induced long intergenic noncoding RNA 1* (*HILinc1*) was identified in pear [239]. The target of this lncRNA binds with HSF to enhance its transcriptional activity, leading to the upregulation of multiprotein bridging factor 1c during heat response.

During cold stress, plants undergo a process known as vernalization, which prevents flowering during vegetative growth in winter and promotes flowering during the reproductive phase under favorable conditions. This regulatory mechanism of vernalization is controlled by FLC. Several lncRNAs, such as COLDAIR [58], COLDWRAP [59], and COOLAIR [60], suppress the expression of FLC through chromatin modification [240]. Furthermore, lncRNAs SVALKA [56], cryptic antisense CBF1 (asCBF1), and CIR (CBF intergenic lncRNA) regulate freezing tolerance through transcription of the CBF1 gene [241]. The COLD INDUCED lncRNA1 (CIL) participates in low-temperature stress response by affecting the reactive oxygen pathway or osmotic adjustment substance [61], while the cis-acting NAT lncRNA_2962 activates transcription of cold-responsive MADS AFFECTING FLOWERING 4 [242]. These lncRNAs affect plant responses to cold by regulating the expression of multiple stress-related genes during the developmental stage.

### 4.5. Combined Stresses

In the face of unexpected climate conditions, plants often undergo simultaneous exposure to multiple stress conditions, which significantly impact their growth, development, and overall yield. Several miRNAs have been identified that target not only transcription factors but also hormone-signaling genes under combined stress conditions. For instance, miR156 has been found to enhance heat and cold stress tolerance in Arabidopsis and rice plants, respectively [204,215]. While in alfalfa, miR156 increased plant tolerance to salt, heat, and drought stress by downregulating the SPL gene [214,243,244]. Another notable example is miR393, which targets the auxin receptors with F-box domains, TIR1 and AFBs, thus forming the miR393-TIR1/AFBs module. This module is highly conserved in many plant species and links auxin response with transcriptional regulation of plants under salt [245] and drought stress conditions [246]. miR394 targets the LEAF CURLING RESPONSIVENESS GENE, an F-box protein (SKP1-Cullin/CDC53-F-box), which regulates drought and salt stress responses through ABA-mediated and CBF-dependent cold acclimation pathways [247,248]. Additionally, miR408 targets both the blue copper protein and laccase genes and ultimately enhances plant tolerance to salt, cold, and oxidative stress [164].

In the context of hormone-mediated responses to combined stress, Qin et al. [171] identified a novel lncRNA, DRIR, that regulates drought and salt stress responses by mediating the expression of ABA signaling genes, stomatal closure, and proline accumulation. Additionally, lncRNA77580 in soybeans modulates the expression of a number of transcription factors under salt and drought stress conditions [249]. Furthermore, using high-throughput sequencing technology, approximately 467 circRNAs were identified in tomatoes in response to drought, heat, and combined stress conditions [250]. Among them, 156 circRNAs were shared among all stress conditions, with circRNA197 and circRNA29 showing significant expression levels in three stress conditions and being predicted to play an important role in photosynthesis and other biological processes. 

Nowadays, deep learning-based methods, such as the PLncPRO (Plant Long Non-Coding RNA Prediction by Random fOrest) tool, are widely used to predict lncRNAs. Singh et al. [22], utilizing this tool, predicted and identified many lncRNAs that were differentially expressed in drought- and salt-tolerant chickpea genotypes at the early and late reproductive stages, respectively. 

## 5. Conclusions and Future Perspectives

Advances in high-throughput sequencing technologies, such as single-cell RNA sequencing, coupled with bioinformatics tools and validatory methods such as northern blot, quantitative real-time PCR, mass spectrometry, and CRISPR/Cas9, have greatly aided the identification of ncRNAs that play a critical role in a variety of biological processes. ncRNAs are dynamic entities with a shared origin and conserved mode of action across different plant species. They exhibit discrete expression patterns across tissues, which can influence growth and development. At the molecular level, multiple regulatory mechanisms, including epigenetic, transcriptional, and post-transcriptional processes, all have an impact on plant performance in response to various abiotic stress conditions. In contemporary research, deep learning methods are frequently employed to predict tissue-specific and environmental-specific novel ncRNAs in many plant species. The lab-based validation of these ncRNAs not only enhances our understanding of their biological function but also equips researchers with tools to harness these molecules for developing plants with improved resilience, yields, and quality in the face of changing climatic conditions. 

**Table 1 ncrna-10-00013-t001:** List of studies that identified ncRNAs (siRNAs, miRNAs and their targets, lncRNAs, and circRNAs) for growth, development, and stress resilience in various plant species.

Trait	siRNAs	Reference
Root development	TAS3-siRNA (ARFs)	Marin et al. [84]
	TAS3-siRNA (leafbladeless1)	Gautam et al. [85]
Leaf development	TAS3-siRNA (ARF3)	Chitwoodet al. [103]
	TAS3-siRNA	Schwab et al. [104]
	TAS3-siRNA (leafbladeless1)	Dotto et al. [105]
Seed, endosperm, nutrient development	maternal p4-siRNAs	Lu et al. [130] Yuan et al. [131]
Drought stress	Dehydration induced siRNAs	Jung et al. [161]
		Yao et al. [162]
Salt stress	OsSIDP301 (stress-induced DUF1644 protein)	Ge et al. [182]
	Response to ABA and Salt 1	Ren et al. [183]
	OsSKL2 and OsASR	Jiang et al. [184]
	NATs of SRO5 and P5CDH	Borsani et al. [185]
Flooding/hypoxia	hypoxia-induced tasiRNA	Moldovan et al. [194]
Heat stress	HEAT-INDUCED TAS1 TARGET1	Li et al. [207]
	Phased siRNA	Pokhrel et al. [208]
Cold stress	cold-induced tasiRNA	Kume et al. [209]
miRNAs and corresponding targets		
Root development	miR165/166-RLD1/2	Gautam et al. [85]
	miR156-SPL10	Barrera-Rojas et al. [86]
	miR165-SHR	Carlsbecker et al. [87]
Leaf development	miR319-TCP	Bresso et al. [106]
	miR390-TAS3	Husbands et al. [108]
	miR396-GRF	Omidbakhshfard et al. [124]
Seed, endosperm, nutrient development	miR156, miR166, miR172, miR319, miR396	Khemka et al. [132]
	miR156-SPL14/SPL16	Miao et al. [133]
Drought stress	miR169-NFYA5	Li et al. [128]
	miR408a-Laccase	Jiao et al. [163]
	miRNA-mediated drought-responsive regulatory network	Balyan et al. [165]
	miR169-NFYA3	Ni et al. [167]
Salt stress	osa-miR12477-LAO	Parmar et al. [180]
Flooding/hypoxia	miR156/SPL	Feyissa et al. [199]
Heat stress	miR164-NAC	Tsai et al. [218]
Cold stress	miR319-TCP	Thiebaut et al. [225]
	miR319-OsPCF6 and OsTCP21	Wang et al. [226]
lncRNAs		
Root development	APOLO	Ariel et al. [89] Moison et al. [91]
	lncWOX11a	Ran et al. [97]
	lncWOX5	Qi et al. [100]
Leaf development	TWISTED LEAF	Liu et al. [110]
	age-related lncRNAs	Kim et al. [111]
Seed, endosperm, nutrient development	LAIR	Wang et al. [134]
Drought stress	DRIR	Qin et al. [171] Dong et al. [172]
Salt stress	Salt-responsive DE-lncRNAs	Mansuri et al. [177] Mansuri et al. [187]
	Genome-wide lncRNAs in *Medicago truncatula*	Wang et al. [188]
Flooding/hypoxia	DE-lncRNAs in response to waterlogging stress	Yu et al. [196] Kęska et al. [202]
Heat stress	TCONS_00048391 and TCONS_00010856	Wang et al. [235]
Cold stress	CIL1	Liu et al. [61]
circRNAs		
Root development	DE-circRNAs	Xu et al. [95]
Leaf development	circ-AT1G29965, circ-AT4G08300, circ-AT5G18590	Liu et al. [112]
Seed, endosperm, nutrient development	DE-circRNAs participate in fatty acid metabolism	Zhou et al. [135]
Drought stress	Circular RNA profiling	Zhang et al. [173]
	circGORK	Ache et al. [174] Becker et al. [175]
Salt stress	circ_000260, circ_001362, and circ_001730 circRNA Chr10:20,345,844|20,346,873	Liu et al. [189] Yin et al. [190]
Heat stress	1583 heat-specific	Pan et al. [212]
Cold stress	163 chilling responsive 1830 low temperatures induced	Zuo et al. [211] Yang et al. [210]

## Figures and Tables

**Figure 1 ncrna-10-00013-f001:**
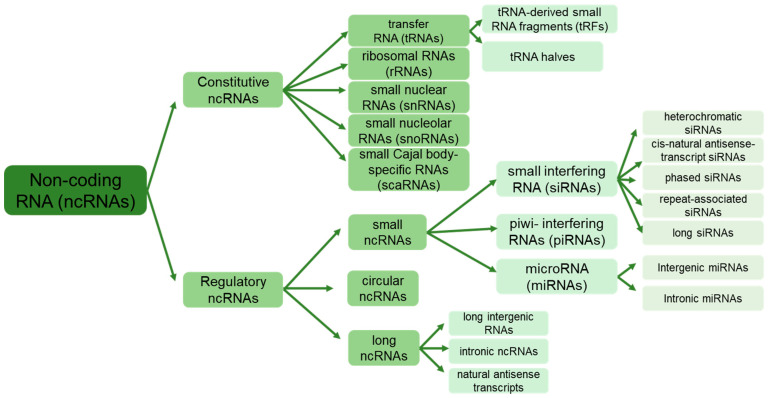
Classification of non-coding RNAs (ncRNAs). Constitutive ncRNAs include transfer, ribosomal, small nuclear, small nucleolar, and small Cajal body-specific RNAs, while regulatory ncRNAs comprise small ncRNAs (small interfering, piwi, and microRNAs), circular ncRNAs, and long ncRNAs.

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
