# Peer review of "The Emerging Role of Non-Coding RNAs (ncRNAs) in Plant Growth, Development, and Stress Response Signaling"

_ncrna, 2024, doi:10.3390/ncrna10010013_

Round 1

Reviewer 1 Report

Comments and Suggestions for Authors

The authors provided a comprehensive review of the functions of ncRNA in plants but lack an explanation of some of the latest developments, such as the discovery of lncRNA-encoded peptides by mass spectrometry, and the new discovery of lncRNA transcription characteristics using single-cell sequencing technology. I think these contents should be reflected in the main text and Future Perspectives section of the paper. In addition, the authors should create a schematic diagram to demonstrate the roles of ncRNAs in plant growth and development, and in various abiotic stress resilience.

Author Response

Response to Reviewer’s comments

Thank you very much for taking the time to review this manuscript. Please find the detailed responses below and the corresponding revisions, corrections, and track changes in the re-submitted files.

Comments 1: The authors provided a comprehensive review of the functions of ncRNA in plants but lack an explanation of some of the latest developments, such as the discovery of lncRNA-encoded peptides by mass spectrometry and the new discovery of lncRNA transcription characteristics using single-cell sequencing technology. I think these contents should be reflected in the main text and Future Perspectives section of the paper.

Response 1: Thank you for bringing these updates to my attention. We agree with this suggestion; therefore, we have introduced cell-type-specific lncRNA and lncRNA-associated gene regulatory networks identified in single-cell RNA sequencing in the revised text. We also introduce the miRNA-encoded peptides and lncRNA-encoded peptides and their role in plant growth and development. Additionally, in the introduction section, we talk about how technologies have been revolutionized by the discoveries of ncRNA. Please see lines (95-111), (209-217), (264-276), and the conclusions and future perspectives section for details about those changes.

Comments 2: In addition, the authors should create a schematic diagram to demonstrate the roles of ncRNAs in plant growth and development, and in various abiotic stress resilience.

Response 2. Thank you for addressing the importance of graphical illustration. In this regard, we have created a schematic diagram (refer to Figure 2) depicting potential non-coding RNAs (ncRNAs) influencing plant root, leaf, seed, and endosperm development, as well as their roles in enhancing resilience to multiple abiotic stresses.

Reviewer 2 Report

Comments and Suggestions for Authors

   In the manuscript entitled ”The emerging role of non-coding RNAs (ncRNAs) in plant  growth, development, and stress response signaling” the authors describe different categories of plant ncRNAs according to their structure and biosynthetic routes. They also provide a kind of “state of the art” regarding the functional implications of ncRNAs in plant physiology.

   In general terms, the information and references mentioned are accurate but, in my opinion, incomplete. It could be enough for the introduction of a dissertation, but not as a general review as it is intended. There are tens of similar reviews in the literature that provide deeper and more recent data.

Some of the contents could be improved/added:

- Information about siRNAs: they are ancestral molecules involved in the recognition of foreign nucleic acids and are known to target transposable elements. Some of them require the action of miRNAs for their biosynthesis (phased siRNAs) and may be involved in plant metabolism.

- Information about miRNAs: the authors could go deeper on it. Furthermore, most of the references in table1 cite studies on Arabidopsis but others involving other cultivated species could be included.

- Information about new classes of ncRNAs, such as tRNA-derived fragments (tRFs) in plants, could be added.

- Information about other kind of stresses, such as the response to heavy metals or low temperatures, could be added.

-In general, authors make reference to studies with single stresses (drought, salt, heat, etc), however, if they want to offer updated information on ncRNAs that help improve resilience of plants to a changing environment,  as they claim, they could also make reference to studies with combined stresses.

Some of the contents should be corrected:

1)      The studies provided as “siRNA” in table 1 do not make reference to real plant siRNAs but to the use of artificial siRNAs to knock-out some genes involved in development or stress responses. It should be modified.

2)      Line 222: ENOD40 instead of ENDO40.

3)      Line 356: “Starch is another the main constituent”

4)      Table 1 include ncRNAs that have been previously mentioned in the manuscript but makes reference to new papers. In general it seems that Table 1 comes from an external source.

According to the literature cited by the authors and the information in the paper it seems to me that their area of expertise are:

1)       lncRNAs:

- Most of the references in the manuscript regarding the involvement of ncRNAs in plant physisology and stress responses (and also when speaking in general terms) rely on this kind of molecules (IncRNA (98 entries); miRNA (45 entries, some of them related to lncRNAs..); siRNA (16 entries)). Other classes of ncRNAs are mentioned but only superficially.

- Furthermore, the lncRNAs described in some genome-wide studies are deeply described in the manuscript (e.g. references 134 and 135 in paragraph 416-424).

2)      Grass species of plants: as most of the examples involve Arabidopsis (as a model plant, with 49 entries), rice (42 entries), maize (17 entries), wheat (14 entries) and other plants are scarcely mentioned.

   In my opinion the manuscript cannot be published in the present form. As I have mentioned before, many general reviews on ncRNAs in plants have been previously/recently published. However, I must recognize that authors seem to possess a deep knowledge in lncRNAs, specially from grass species. For that reason I would like to encourage authors to reformat the paper to provide a more humble and specific/focused view that, in my opinion, could be new and more interesting to some readers.

Author Response

Thank you very much for taking the time to review this manuscript. Please find the detailed responses below and the corresponding revisions, corrections, and track changes in the re-submitted files.

In the manuscript entitled” The emerging role of non-coding RNAs (ncRNAs) in plant  growth, development, and stress response signaling” the authors describe different categories of plant ncRNAs according to their structure and biosynthetic routes. They also provide a kind of “state of the art” regarding the functional implications of ncRNAs in plant physiology.

In general terms, the information and references mentioned are accurate but, in my opinion, incomplete. It could be enough for the introduction of a dissertation, but not as a general review as it is intended. There are tens of similar reviews in the literature that provide deeper and more recent data.

Some of the contents could be improved/added:

- Information about siRNAs: they are ancestral molecules involved in the recognition of foreign nucleic acids and are known to target transposable elements. Some of them require the action of miRNAs for their biosynthesis (phased siRNAs) and may be involved in plant metabolism.

Response: Thank you for bringing these updates to my attention. We have improved this section by adding different types of siRNA (exo and endo), their biogenesis, and a brief explanation of their regulatory role in the biological processes of plants. (Line#158-187)

- Information about miRNAs: the authors could go deeper on it. Furthermore, most of the references in table1 cite studies on Arabidopsis but others involving other cultivated species could be included.

Response: Thank you for bringing these updates to our attention. We have improved this section by incorporating more information on miRNA biogenesis, miRNA-encoded peptides, and their regulatory function in plant development. Additionally, we have included additional studies from different plant species in Table 1. (Line# 195-217)

- Information about new classes of ncRNAs, such as tRNA-derived fragments (tRFs) in plants, could be added.

Response: Thank you for bringing it to our attention. We have included a section about tRNA-derived fragments (tRFs), biogenesis and its regulatory function in plant development. (Line# 128-140)

- Information about other kind of stresses, such as the response to heavy metals or low temperatures, could be added.

Response: Thank you for bringing it to our attention. We have introduced the role of ncRNA for high and low temperature stresses. (Line# 704-749)

-In general, authors make reference to studies with single stresses (drought, salt, heat, etc), however, if they want to offer updated information on ncRNAs that help improve resilience of plants to a changing environment, as they claim, they could also make reference to studies with combined stresses.

Response: Thank you for bringing it to our attention. We have introduced the role of ncRNA under combined stress conditions. (Line# 751-785)

Some of the contents should be corrected:

  • The studies provided as “siRNA” in table 1 do not make reference to real plant siRNAs but to the use of artificial siRNAs to knock-out some genes involved in development or stress responses. It should be modified.

Response: Thank you for bringing it to our attention. We have introduced real plant siRNA involved in plant growth and development, as shown in Table 1.

  • Line 222: ENOD40 instead of ENDO40.

Response: Thank you for bringing it to our attention. We have corrected this mistake in the text.

  • Line 356: “Starch is another the main constituent”

Response: Thank you for bringing it to our attention. We have reframed this sentence in the text.

  • Table 1 include ncRNAs that have been previously mentioned in the manuscript but makes reference to new papers. In general it seems that Table 1 comes from an external source.

Response: Thank you for bringing this to our attention. We have updated the table and incorporated the references into the main text.

Round 2

Reviewer 1 Report

Comments and Suggestions for Authors

The authors have already answered my questions, I have no more questions.

Author Response

We would like to thank both the anonymous reviewers for providing valuable suggestions and insightful comments on this review article. Please find the detailed responses below, along with revisions and corrections, in the resubmitted file.

Reviewer 1: The authors have already answered my questions; I have no more questions.

Response: We appreciate your thorough review of this article. We would like to mention that, since the figure serves as a representation of Table 1, we removed it from the revised version of the manuscript to avoid duplication.

Reviewer 2 Report

Comments and Suggestions for Authors

Dear authors of the manuscript “The emerging role of non-coding RNAs (ncRNAs) in plant growth, development, and stress response signaling”.

I am pleased to know that you have made a considerable effort to improve the initial manuscript. However, there are still important conceptual errors that must be corrected before publication.

1)    Table 1:

I have detected some inaccuracies in Table 1, particularly in the “siRNA” column, with data such as “AGO1”, “AGO10”, “RDR”, “DCL”… As you know these are real proteins involved in the biosynthesis of some ncRNAs and also in the gene silencing complexes related to the post-transcriptional activity of those ncRNAs. However, they’re not siRNAs, neither are the table entries with the values “DSG”, “RAS1…

As an example, the authors of the bibliographic reference number 85 (Sorin et al…) use AGO1 mutants to show the relevance of transcriptional silencing in root development. However, there’s no reference to any siRNA in it! and AGO1 is NOT an siRNA!..

Something similar happens with reference number 104 (Liu et al.) with AGO10 mutants…And so on with other references…These are serious mistakes that cannot be tolerated in a review manuscript.

On the other hand, regarding the miRNA data, it should be explained in the table that the name of the miRNA (e.g. miR319) and its putative target (e.g. TCP) are both included in the entry of the table “miR319-TCP”.

In general terms, the contents in Table 1 should be modified/corrected/improved before publication.

Furthermore, Figure 2 is redundant, as it does not provide additional information to that already described in Table 1, while containing the same inaccuracies… Only one should be kept in the manuscript.

2)    Paragraph 306-310 should be rewritten it seems to be incomplete. Please, note that LHP1 belongs to the polycomb repressive complex 1.

3)    Line 441: It is not explained what “SBEIIb” is.

4)    The references in the bibliography numbered as 145 and 154 are incomplete.

Author Response

We would like to thank both the anonymous reviewers for providing valuable suggestions and insightful comments on this review article. Please find the detailed responses below, along with revisions and corrections, in the resubmitted file.

Reviewer 2: I am pleased to know that you have made a considerable effort to improve the initial manuscript. However, there are still important conceptual errors that must be corrected before publication.

1)    Table 1:

I have detected some inaccuracies in Table 1, particularly in the “siRNA” column, with data such as “AGO1”, “AGO10”, “RDR”, “DCL”… As you know these are real proteins involved in the biosynthesis of some ncRNAs and also in the gene silencing complexes related to the post-transcriptional activity of those ncRNAs. However, they’re not siRNAs, neither are the table entries with the values “DSG”, “RAS1…

As an example, the authors of the bibliographic reference number 85 (Sorin et al…) use AGO1 mutants to show the relevance of transcriptional silencing in root development. However, there’s no reference to any siRNA in it! and AGO1 is NOT an siRNA!..

Something similar happens with reference number 104 (Liu et al.) with AGO10 mutants…And so on with other references…These are serious mistakes that cannot be tolerated in a review manuscript.

On the other hand, regarding the miRNA data, it should be explained in the table that the name of the miRNA (e.g. miR319) and its putative target (e.g. TCP) are both included in the entry of the table “miR319-TCP”.

In general terms, the contents in Table 1 should be modified/corrected/improved before publication.

Response: Thank you for bringing these points to our attention. We have improved Table 1 by including real siRNAs for various biological processes. Additionally, we have mentioned in the table legends that miRNAs are shown with their corresponding targets.

Furthermore, Figure 2 is redundant, as it does not provide additional information to that already described in Table 1, while containing the same inaccuracies… Only one should be kept in the manuscript.

Response: Thank you for bringing this to our attention. Yes, indeed, there should be only one either table or figure. We have removed Figure 2 from the revised version.

2)    Paragraph 306-310 should be rewritten it seems to be incomplete. Please, note that LHP1 belongs to the polycomb repressive complex 1.

Response: Thank you for bringing these issues to our attention. We have rewritten this section to ensure that the information flows smoothly.

3)    Line 441: It is not explained what “SBEIIb” is.

Response: Thank you for bringing this to our attention. We have abbreviated "SBEIIb" in the text section.

4)    The references in the bibliography numbered as 145 and 154 are incomplete.

Response: Thank you for bringing this to our attention. We have updated the incomplete and newly added references.

Round 3

Reviewer 2 Report

Comments and Suggestions for Authors

Dear authors,

Thanks for attending my suggestions and sorry for all the inconveniences I may have caused. I hope you too feel satisfied with the results.

Best wishes,